# Ebola and Marburg virus matrix layers are locally ordered assemblies of VP40 dimers

William Wan[1,2†], Mairi Clarke[1], Michael J Norris[3], Larissa Kolesnikova[4], Alexander Koehler[4], Zachary A Bornholdt[5‡], Stephan Becker[4], Erica Ollmann Saphire[3], John AG Briggs[1,6*]

[1]Structural and Computational Biology Unit, European Molecular Biology Laboratory, Heidelberg, Germany; [2]Department of Molecular Structural Biology, Max Planck Institute of Biochemistry, Martinsried, Germany; [3]Center for Infectious Disease and Vaccine Research, La Jolla Institute for Immunology, La Jolla, United States; [4]Institut für Virologie, Philipps-Universität Marburg, Hans-Meerwein-Straße, Marburg, Germany; [5]The Scripps Research Institute, La Jolla, United States; [6]Structural Studies Division, MRC Laboratory of Molecular Biology, Cambridge, United Kingdom

*For correspondence:
jbriggs@mrc-lmb.cam.ac.uk

Present address: [†]Department of Biochemistry and Center for Structural Biology Vanderbilt University, Nashville, United States; [‡]Mapp Biopharmaceutical, Inc, San Diego, United States

Competing interests: The authors declare that no competing interests exist.

**Abstract** Filoviruses such as Ebola and Marburg virus bud from the host membrane as enveloped virions. This process is achieved by the matrix protein VP40. When expressed alone, VP40 induces budding of filamentous virus-like particles, suggesting that localization to the plasma membrane, oligomerization into a matrix layer, and generation of membrane curvature are intrinsic properties of VP40. There has been no direct information on the structure of VP40 matrix layers within viruses or virus-like particles. We present structures of Ebola and Marburg VP40 matrix layers in intact virus-like particles, and within intact Marburg viruses. VP40 dimers assemble extended chains via C-terminal domain interactions. These chains stack to form 2D matrix lattices below the membrane surface. These lattices form a patchwork assembly across the membrane and suggesting that assembly may begin at multiple points. Our observations define the structure and arrangement of the matrix protein layer that mediates formation of filovirus particles.

## Introduction

The filovirus family includes viruses such as Ebola, Marburg, and Sudan viruses that can cause hemorrhagic fever and severe disease (*Feldmann et al., 2013*). Filoviruses package their single-stranded negative-sense RNA genomes with viral proteins including nucleoprotein (NP), VP24 and VP35, into helical ribonucleoprotein assemblies called nucleocapsids (NCs) (*Huang et al., 2002*; *Noda et al., 2010*; *Wan et al., 2017*). NCs are recruited to the plasma membrane and bud from the host cells as enveloped virions with a characteristic filamentous morphology from which the family takes its name (*Geisbert and Jahrling, 1995*).

The filovirus matrix protein, VP40, binds to and concentrates at the plasma membrane of infected cells, where it can interact with components of the NC to promote envelopment, and where it drives formation of the filamentous virus particles. VP40 is required for viral budding, and expression of VP40 alone is sufficient to drive formation of filamentous virus-like particles (VLPs) containing a matrix layer and membrane envelope (*Harty et al., 2000*; *Jasenosky et al., 2001*; *Noda et al., 2002*; *Timmins et al., 2001*). The morphology of these VLPs is similar to that of true virions but their diameter is smaller (*Noda et al., 2002*). When VP40 is co-expressed with NC components NP, VP24

and VP35, VLPs are produced which are almost indistinguishable from true virions (*Bharat et al., 2012*; *Noda et al., 2005*; *Wan et al., 2017*).

A number of crystal structures have been determined of Ebola virus VP40 (eVP40) and Sudan (Ebola) virus (sVP40) (*Bornholdt et al., 2013*; *Clifton et al., 2015*; *Dessen et al., 2000*; *Gomis-Rüth et al., 2003*). VP40 contains an N-terminal domain (NTD) and a C-terminal domain (CTD), linked by an intrasubunit hinge. Both eVP40 and sVP40 have been crystallized in space group C2 with similar unit cell dimensions. These crystals reveal dimers assembled via a hydrophobic interface in the NTD, involving residues A55, H61, F108, A113, M116, and L117 which are distributed across two alpha-helices (residues 52–65, 108–117). Disruption of the NTD dimer interface by site-directed mutagenesis prevents migration of VP40 to the plasma membrane and prevents matrix assembly (*Bornholdt et al., 2013*). Within the typical C2 crystal packing of unmodified VP40, eVP40, and sVP40 dimers are further arranged in linear assemblies via a hydrophobic CTD-CTD interface (*Bornholdt et al., 2013*; *Dessen et al., 2000*; *Figure 1—figure supplement 1A,B*). This interface involves residues L203, I237, M241, M305, and I307, which together form a relatively smooth hydrophobic patch (*Figure 1—figure supplement 1A*). The CTD also contains a basic patch composed of six lysine residues (K221, K224, K225, K270,K274, K275), which is essential for matrix assembly and membrane budding (*Figure 1—figure supplement 1A*).

A crystal structure of Marburg VP40 (mVP40) has also been determined (*Oda et al., 2016*; *Figure 1—figure supplement 1C*). eVP40 and mVP40 have 42% sequence identity and a C-alpha RMSD of 2.4 Å in the NTD, but only 16% sequence identity and 5.6 Å C-alpha RMSD in the CTD. The overall topology of mVP40, however, is similar to that of eVP40. The mVP40 monomer has similar N- and C-terminal domains, although there is a small rotation of the CTD relative to the NTD when compared to eVP40. mVP40 dimerizes via an NTD interface that is very similar to that in eVP40, and mVP40 also forms dimers in solution that are required for membrane binding and filament budding. The mVP40 CTD basic patch is also required for membrane binding but is larger and flatter than that in eVP40. In the mVP40 crystal packing, the CTDs meet at an angle and do not form the more extensive hydrophobic interface observed in the C2 crystals of eVP40 and sVP40 (*Figure 1—figure supplement 1*).

Deletion or proteolysis of the C-terminus or C-terminal domain or incubation with urea drives oligomerization of the eVP40 NTD into RNA-binding octameric rings (*Bornholdt et al., 2013*; *Gomis-Rüth et al., 2003*). Subsequent work suggests that octameric rings are likely to have a function during the viral lifecycle independent of matrix formation (*Bornholdt et al., 2013*; *Gomis-Rüth et al., 2003*; *Hoenen et al., 2005*; *Hoenen et al., 2010b*).

In an effort to mimic membrane-associated electrostatic conditions, a crystal structure of eVP40 was determined in the presence of the negatively charged additive dextran sulfate (*Bornholdt et al., 2013*). Under these conditions, eVP40 assembled into linear hexamers (*Figure 1—figure supplement 1D*) with unit distances approximately consistent with earlier, lower resolution tomographic analysis of the VP40 layer in Ebola and Marburg virions (*Beniac et al., 2012*; *Bharat et al., 2011*). The core of the linear hexamer consists of four NTDs from which the linked CTDs are disordered or 'sprung' and not resolved. The first and sixth VP40s in the hexamer retain their CTD in close association with its NTD. These CTDs assemble into linear filaments via the same CTD-CTD interactions observed in the C2 crystals of VP40 dimers. The NTD-NTD interfaces within the hexamer alternate between the dimer interface and the same NTD-NTD interface observed in the octameric ring. No linear hexamer structure has been determined for mVP40. However, mutagenesis of residues in mVP40 homologous to those forming the 'octamer-like' interfaces in hexameric eVP40 (*Bornholdt et al., 2013*; *Hoenen et al., 2010a*) retains the ability of mVP40 to dimerize but prevents membrane binding and budding. Based on existing data it seems likely that VP40 is arranged in a similar way in both MARV and EBOV particles.

The current model for the assembly state of VP40 within filovirus particles consists of VP40 hexamers as crystallized in the presence of dextran sulfate, arranged to form a 2D lattice, with dimensions of the 2D lattices in the model based upon repeating features observed in low-resolution cryo-electron tomography (cryo-ET) studies (*Beniac et al., 2012*; *Bharat et al., 2011*; *Bornholdt et al., 2013*). Analysis of transfected cells suggested that VP40 assembles into hexamers and octamers at the plasma membrane and in protrusions (*Adu-Gyamfi et al., 2012*) and that the interaction is mediated by the C-terminal domains (*Adu-Gyamfi et al., 2013*). The structure and arrangement of VP40 within actual assembled virus particles, however, is unknown. It therefore remains unclear how VP40

assembles in the actual virion, which model of VP40 assembly best reflects that in the virion, and how VP40 induces membrane curvature and assembly with other viral components. Here, we have set out to directly determine the structure and arrangement of VP40 within filamentous virus-like particles and authentic filovirus virions.

## Results

The linear CTD-CTD interface is consistently observed in unmodified VP40 crystals eVP40 and sVP40, with intact CTDs and in the absence of charged additives, consistently crystallize in linear filaments of dimers in the space group C2 (*Bornholdt et al., 2013*; *Clifton et al., 2015*; *Dessen et al., 2000*). In an attempt to determine if this linear arrangement is an inherent preferred assembly interface of eVP40 or simply the result of the common C2 crystal packing, we crystallized eVP40 in two alternate crystal forms: $P6_2$ and $P6_422$. Notably, in both of these crystal forms, eVP40 also builds linear filaments of dimers, mediated by CTD-CTD interdimer interfaces, with CTD basic patches displayed on a common face. These filaments differ from the C2 filaments by slight torsional rotation about the relatively flat hydrophobic CTD-CTD interface (*Figure 1—figure supplement 1E,F* and *Table 1*). The propensity of VP40 to form linear assemblies by CTD-CTD interactions across multiple crystal forms suggest this is a biologically preferred interface and may be important in the viral particle or virus assembly.

The structure of the matrix layer in EBOV VLPs eVP40 expression induces budding of long VLPs from the surface of mammalian cells (*Noda et al., 2002*; *Timmins et al., 2001*). We purified Zaire eVP40 VLPs by sucrose gradient purification and imaged them by cryo-ET, finding multi-micron long filaments with a diameter of ~ 28 nm (*Table 2*) and a matrix-like protein layer visible under the membrane bilayer. We applied subtomogram averaging methods to determine the structure of the matrix layer to a resolution of 10 Å (*Figure 1*, *Figure 1—figure supplement 2*) from intact eVP40 VLPs. We observed that the matrix layer is formed by higher-order linear oligomerization of VP40 dimers on the inner surface of the viral membrane. VP40 dimers form long chains that stack to form 2D lattices with a monoclinic p2 space group in the plane of the membrane (*Figure 1*, *Table 2*). The crystal structure of the C2 eVP40 dimer (PDB: 4LDB) could be fit as a rigid body into the density, showing that linear oligomerization is mediated by CTD to CTD interactions.

In addition to eVP40 VLPs, we also produced VLPs by co-expression of eVP40 with the Ebola virus glycoprotein GP, and by co-expression of eVP40 with the NC components NP, VP24 and VP35. As described previously (*Bharat et al., 2012*), NP-VP24-VP35-VP40 VLPs have substantially wider filaments to accommodate the NC-like assembly (*Table 2*). We determined the structures of eVP40 within these VLPs at resolutions of 10 Å, (*Figure 1*, *Figure 1—figure supplement 2*). As in the eVP40 VLPs, the matrix layer in these VLPs is formed from extended chains of eVP40 that stack to form monoclinic p2 lattices (*Figure 1*, *Table 2*).

### EBOV matrix layer is formed by oligomerization of VP40 dimers into chains

We were able to fit the C2 crystallographic eVP40 dimer as a rigid body into the matrix structures from all three VLPs (*Figure 1*). We did not observe any substantial electron density that is not occupied by eVP40. The eVP40 dimers are oriented similarly such that the basic patches in the CTDs all point toward and contact the membrane. This orientation is consistent with previous studies, which showed that mutations within this patch modulate membrane binding (*Bornholdt et al., 2013*), and consistent with linear assemblies observed in crystals in which the basic patches are oriented in the same direction.

In all VLPs, oligomerization of eVP40 dimers to form extended chains occurs through a hydrophobic surface patch in the CTD. A hydrophobic CTD-CTD interaction is also found in each of the C2, $P6_2$ and $P6_422$ eVP40 and sVP40 crystal forms, with slightly varying orientations about the CTD-CTD interface (*Figure 1—figure supplement 1*, *Figure 1—figure supplement 3*). In contrast to the flexible CTD-CTD interface, the dimeric NTD-NTD interfaces are considered rigid (*Bornholdt et al., 2013*), and are largely conserved in different crystal structures (*Figure 1—figure supplement 3*). The flexibility about the CTD-CTD interfaces, and possibly also from the intrasubunit NTD-CTD hinge, appear sufficient to accommodate the varying radii of assembled matrix layers.

**Table 1.** Crystallographic data collection and refinement statistics.

| | eVP40 P6$_2$ (7JZJ) | eVP40 P6$_4$22 (7JZT) |
|---|---|---|
| *Data collection* | | |
| Space group | P6$_2$ | P6$_4$22 |
| Unit cell dimensions | | |
| a, b, c (Å) | 159.94, 159.94, 89.75 | 105.28, 105.28, 463.74 |
| α, β, γ (°) | 90, 90, 120 | 90, 90, 120 |
| Wavelength (Å) | 0.9795 | 0.9793 |
| Resolution range (Å)* | 79.97–2.46 (2.59–2.46) | 19.87–4.78 (4.86–4.78) |
| Observations* | 350132 (55437) | 95505 (4826) |
| Unique reflections* | 44360 (6904) | 12595 (607) |
| Completeness (%)* | 93.1 (99.8) | 99.5 (100) |
| Redundancy* | 7.9 (8.0) | 12.1 (13) |
| CC$_{1/2}$* | 0.999 (0.443) | 0.99 (0.60) |
| I/σI* | 15.3 (0.90) | 7.5 (0.3) |
| R$_{merge}$* | 0.092 (3.204) | 0.211 (8.301) |
| R$_{pim}$* | 0.052 (1.795) | 0.081 (3.187) |
| *Anisotropy correction* | | |
| Anisotropic resolution (Å) | | 5.54 (0.89 a* - 0.45 b*) |
| (direction) | | 5.54 b* |
| | | 3.60 c* |
| Resolution after correction* | | 19.88–3.77 (4.30–3.77) |
| No. of unique reflections* (ellipsoidal) | | 6953 (376) |
| I/σI (ellipsoidal)* | | 12.3 (1.8) |
| Completeness (ellipsoidal) (%)* | | 90.8 (64.3) |
| *Refinement* | | |
| No. of atoms | 7452 | 6795 |
| R$_{cryst}$/R$_{free}$ (%) | 24.7/25.5 | 31.5/34.5 |
| Ramachadran plot | | |
| Outliers (%) | 0.00 | 0.23 |
| Allowed (%) | 1.15 | 4.74 |
| Favored (%) | 98.85 | 95.03 |
| RMSD from ideal geometry | | |
| Bond length (Å) | 0.013 | 0.005 |
| Bond angles (°) | 1.96 | 1.05 |
| Clashscore | 1.52 | 8.99 |
| Average B factor | 89.32 | 185.96 |
| Refinement program | Phenix | Phenix |

*Numbers in parentheses correspond to the outer resolution shell.

We previously probed the CTD-CTD interface by introducing either an M241R point mutation or an I307R mutation, both of which lie in the CTD-CTD interface. I307R was combined with R134A in the octameric assembly site to inhibit octamer formation. VP40 mutants bearing M241R or I307R substitutions do not assemble VLPs. We sought to identify an alternate mutation that would stabilize, instead of disrupt, the CTD-CTD interaction. We generated eVP40 bearing an M305F/I307F double mutation, which modeling studies suggested would support hydrophobic packing at the

**Table 2.** Unit cell and filament dimensions matrix layers.

| Specimen | Radius (nm) | a (Å) | b (Å) | θ (°) | α (°) |
| --- | --- | --- | --- | --- | --- |
| Ebola VP40 VLPs | 28 ± 6 (n = 42) | 82 | 47 | 80 | 35 |
| Ebola VP40-GP VLPs | 25 ± 3 (n = 60) | 80 | 48 | 81 | 42 |
| Ebola NP-VP24-VP35-VP40 VLPs | 41 ± 2 (n = 54) | 81 | 50 | 84 | 29 |
| Marburg Virus | 43 ± 2 (n = 75) | 78 | 60 | 54 | -1 |
| Marburg VP40 VLPs | 25 ± 2 (n = 25) | 83 | 46 | 77 | 35 |

Unit cell dimensions are illustrated in *Figures 1* and *2* and are defined as follows: a is the distance between VP40 dimers along the chains, b is the distance between dimers between chains, θ is the internal angle of the lattice, and α is the rotational angle of the unit cell.

interface (*Figure 1—figure supplement 4*). Although eVP40 M305F/I307F overall expressed to a lower yield, the relative proportion of VLP budding was enhanced over wild-type (*Figure 1—figure supplement 4*).

## The structure of the matrix layer in MARV

In order to determine if the disparate sequence of the MARV VP40 CTD still resulted in a matrix assembly similar to that of eVP40, we prepared and purified mVP40 VLPs and determined the structure of the matrix layers to 10 Å resolution (*Figure 2*, *Figure 2—figure supplement 1*). The matrix layer appears similar to that seen in eVP40 VLPs, adopting a p2 lattice with similar dimensions (*Figures 1* and *2*), suggesting that the structure is conserved despite sequence divergence (34% identical, 49% homologous). We fit the dimeric mVP40 crystal structure as well as a dimeric eVP40 crystal structure into the density as a rigid body. For mVP40, there were clashes of the CTDs at the interdimer interface (*Figure 2—figure supplement 2*), while eVP40 fit these densities well. This suggests to us that the CTD of mVP40 is rotated slightly about the CTD-NTD hinge into a position more similar to that of eVP40 when assembled in VP40 VLPs. A structural change in mVP40 in membrane binding has been proposed in prior simulation studies (*Bhattarai et al., 2017*).

We generated authentic MARV virions by infection of Huh7 cells and imaged fixed, purified virions by cryo-ET. MARV virion matrix layers again consist of VP40 dimers forming extended chains through their CTDs and stacking of these chains form a 2D p2 lattice (*Figure 2*), but the lattice angles differ from those in VLPs: the VP40 chains run nearly perpendicular to the filament axis, and the register of neighboring chains differs by approximately half a VP40 protomer from those seen in VLPs (*Figures 1* and *2*). Rigid body fitting of mVP40 dimers or eVP40 dimers shows a good fit with no extra, unassigned densities (*Figure 2*), suggesting that VP40 is the only component in the matrix layer. At this resolution we are unable to confidently assess whether the CTD has rotated slightly relative to the NTD or not.

We attempted to determine the structure of the matrix layer within authentic, fixed Ebola virions, but found that some membranes were 'moth-eaten', leaving membrane and matrix layers disrupted, while in others there were only few places where an ordered matrix layer was observed (*Figure 3—figure supplement 1*). We were therefore unable to determine a structure for VP40 within authentic EBOV virions.

## Global order of filovirus matrices

Lines of density that correspond to VP40 dimer chains are directly visible in tomograms of VLPs and viruses (*Figure 3A*). General features such as the orientation of the chains relative to the axis of the filamentous particle are consistent with those determined by subtomogram averages.

When determining structures, subtomogram averaging provides the position and orientation of each VP40 dimer-centered subtomogram within the tomogram. Visualizations of these positions and orientations are called 'lattice maps' and reveal the global arrangement of VP40 (*Figure 3B,C*). Lattice maps show that in all VLPs studied, 2D lattices form locally ordered patches, and that there are disordered areas or other defects in crystallographic packing between the patches. The local pitch of the array is somewhat variable, and VP40 chains can terminate and run into each other. The overall topology of the matrix layer is a 'patchwork' of locally ordered 2D lattices.

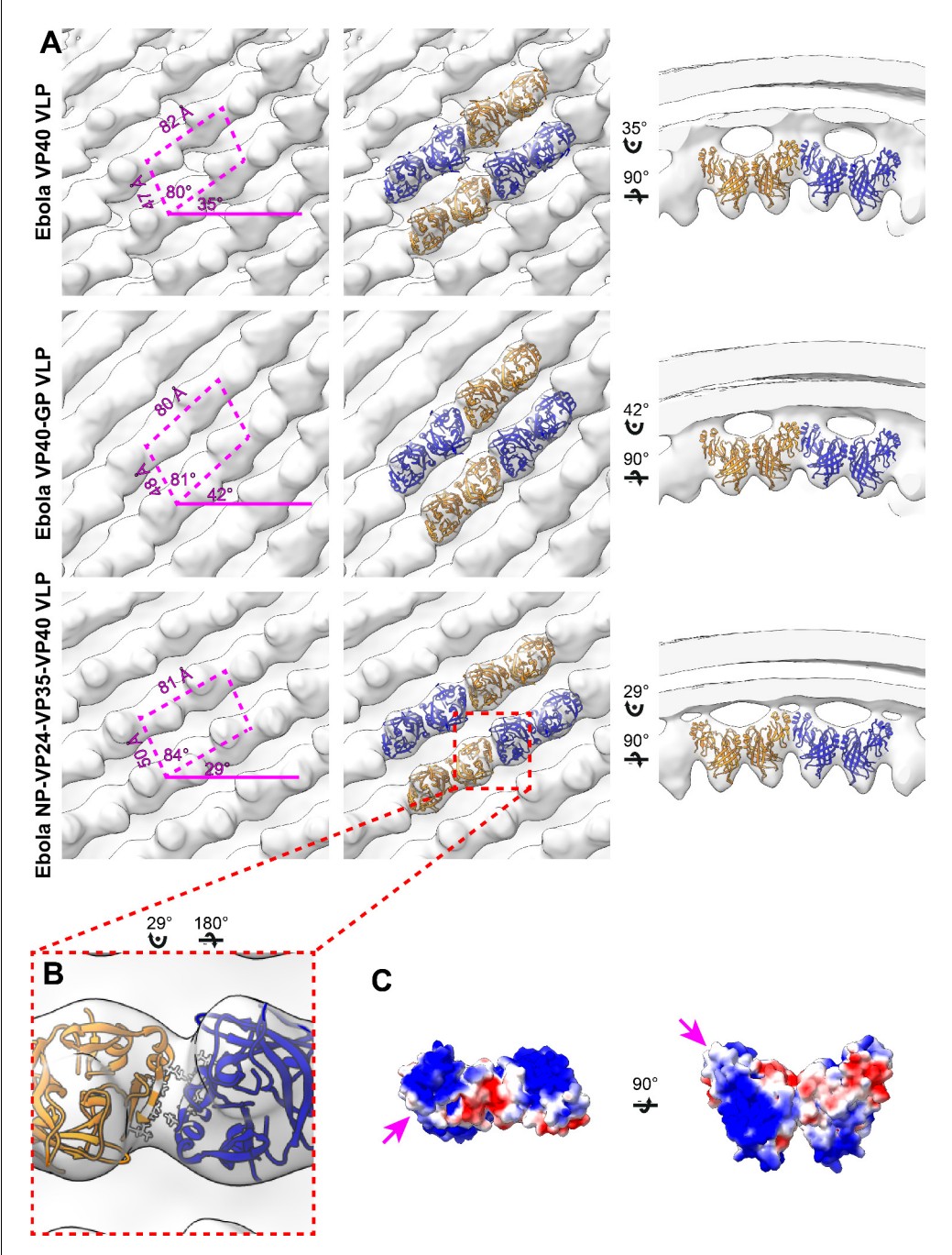

**Figure 1.** Subtomogram averages of the eVP40 matrix layer in VLPs. (**A**) The structure of the matrix layer in Ebola VP40, VP40-GP, and NP-VP24-VP35-VP40 VLPs. For these rows, the left column shows a portion of the subtomogram average from within the VLP; overlaid are the approximate unit-cell dimensions of the 2D lattice. eVP40 dimers are fitted as rigid bodies in the central column (PDB: 4ldb). The right column shows a cross-sectional view parallel to a VP40 linear chain. (**B**) A detailed view of the inter-dimeric CTD-CTD interface in Ebola NP-VP24-VP35-VP40 matrix, with hydrophobic residues at the inter-dimer interface shown in white; this interface is present in all three VLPs. (**C**) Electrostatic maps of the eVP40 dimer, with the hydrophobic patch forming the inter-dimer interface marked by an arrowhead.

The online version of this article includes the following figure supplement(s) for figure 1:

**Figure supplement 1.** Comparison of eVP40 assembly models.
**Figure supplement 2.** Fourier shell correlations of eVP40 subtomogram averages.
**Figure supplement 3.** Comparison of crystal packings observed in eVP40 structures.
**Figure supplement 4.** Characterization of mutations that stabilize the CTD-CTD interface.

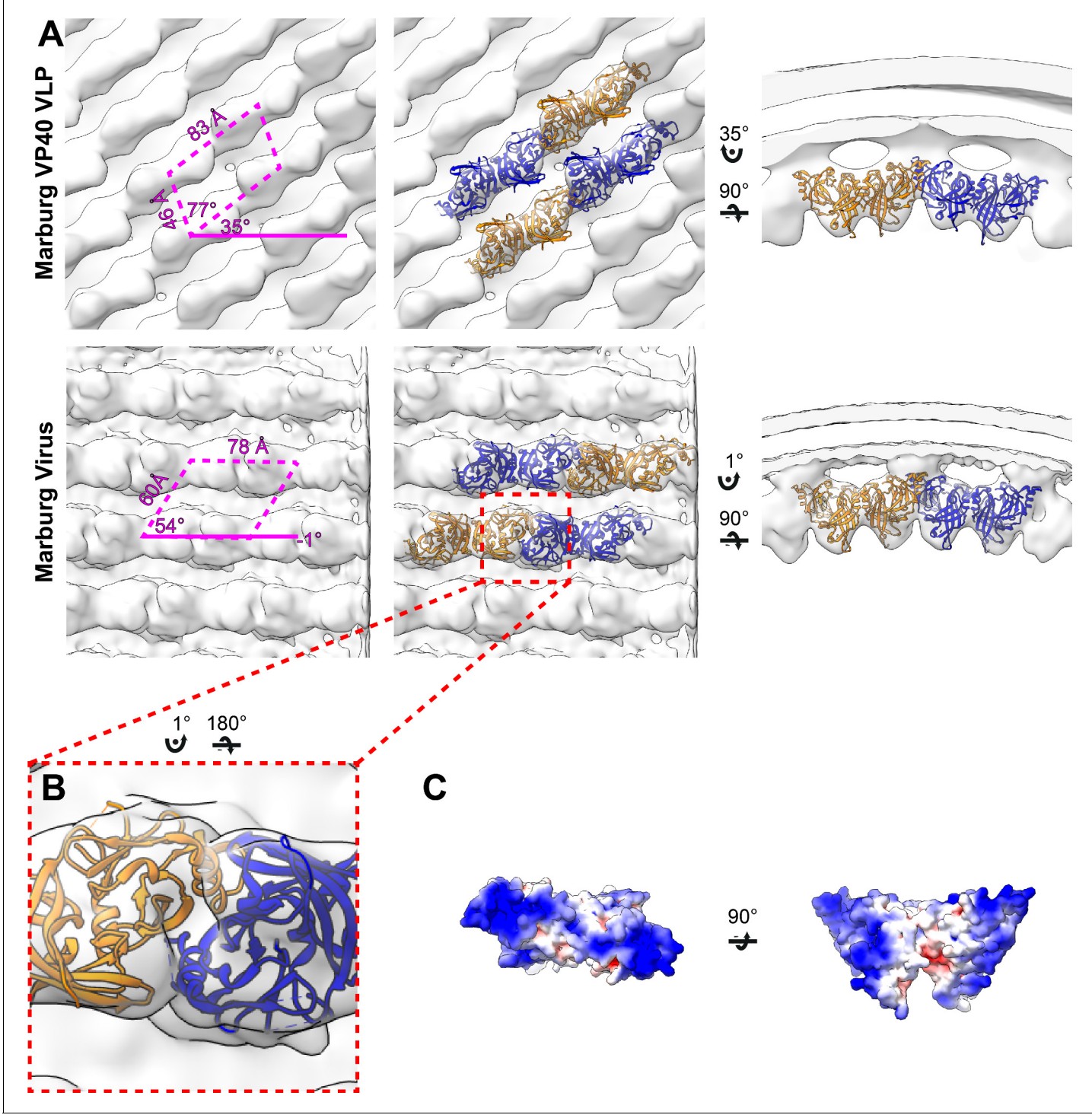

**Figure 2.** Subtomogram averages of the mVP40 matrix layer in VLPs and virions. (**A**) Top row shows the structure of the matrix layer in Marburg VP40 VLPs and bottom row shows Marburg virus. Left column shows a portion of the subtomogram average from within the filaments; overlaid are the approximate unit-cell dimensions of the 2D lattice. Center column shows the same view, but rigid-body fitted mVP40 dimers (PDB: 5b0v). Right column shows the same rigid-body fitting as in the center column, but as a cross-sectional view parallel to a VP40 linear chain. (**B**) A detailed view of the inter-dimeric CTD-CTD interface. (**C**) Electrostatic maps of mVP40 dimer.

The online version of this article includes the following figure supplement(s) for figure 2:

**Figure supplement 1.** Fourier shell correlations of mVP40 subtomogram averages.

**Figure supplement 2.** Rigid body fitting of crystal structures into mVP40 matrix layers.

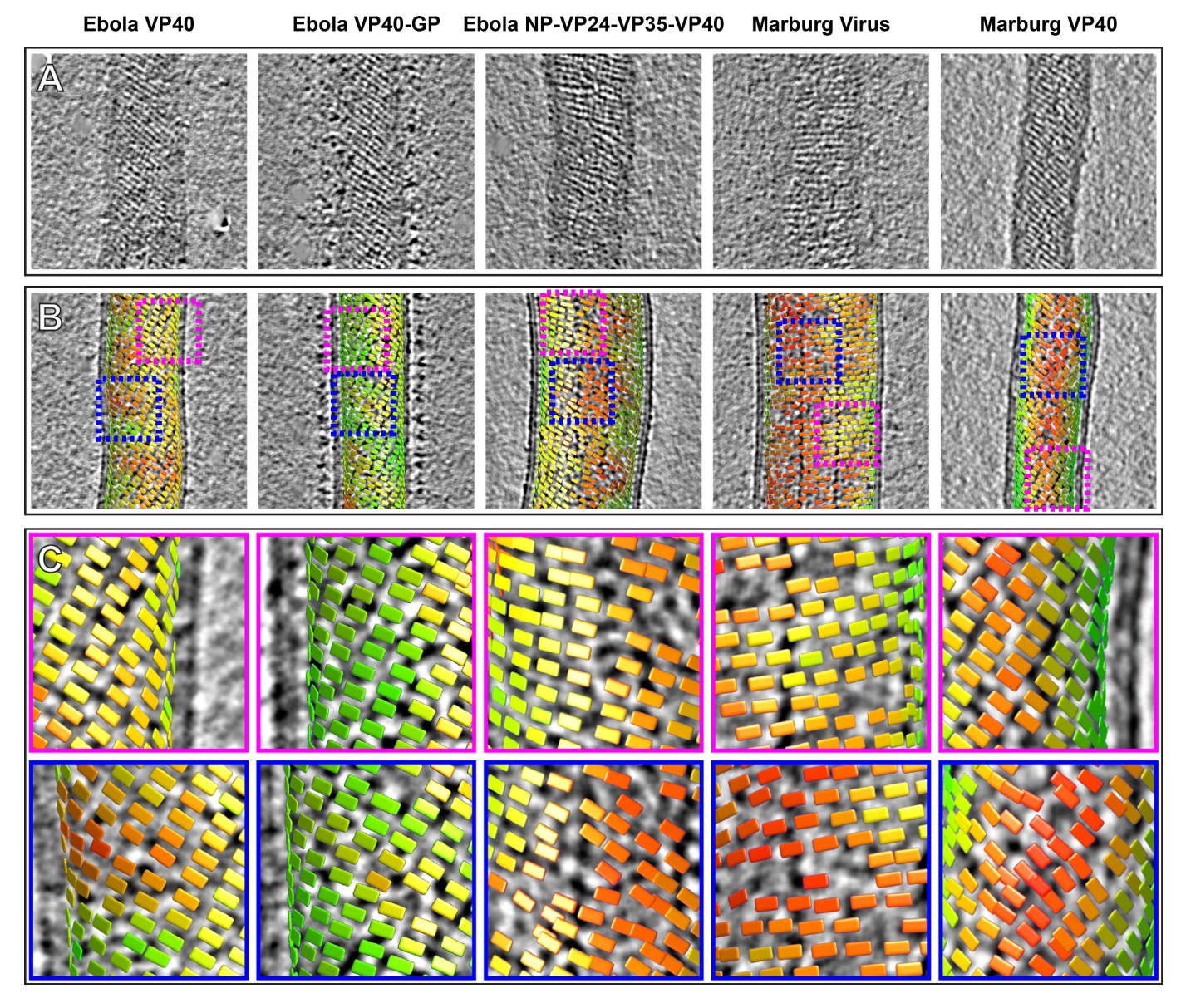

**Figure 3.** Tomographic slices and lattice maps of matrix layers. (**A**) Tomographic slice of the matrix layer protein density directly under the membrane bilayer; VP40 chains are seen as dark lines of density. (**B**) Central slice through each filament with lattice maps overlaid. VP40 dimers are visualized as rectangles, which connect into linear chains along the short sides. Colors are scaled from green to red, which denote high and low correlation scores, respectively. Low correlation scores are generally associated with regions where the local lattice is broken, thus correlating poorly with the strong lattice in the subtomogram average. Representative well-ordered regions are boxed in magenta, while representative poorly ordered regions are boxed in blue. Both are shown in detail in **C**.

The online version of this article includes the following figure supplement(s) for figure 3:

**Figure supplement 1.** Tomographic slices of Ebola virus matrix layers.

Based on the relative positions of subtomograms as visualized in the lattice maps, we calculated and plotted the average radius for each filamentous particle (at the matrix layer) against the pitch angle of the VP40 chains relative to the circumference of the VLP (*Figure 4*). We find that despite the large differences in radius and angle, the radius of curvature of the VP40 chains is similar in all VLPs. Because we had not determined the structure, we did not derive these parameters for Ebola virions. Nevertheless, where small ordered regions of VP40, or isolated VP40 chains were observed,

they had a variable, but small angle relative to the filament, suggesting they have a radius of curvature similar to that observed in NP-VP24-VP35-VP40 VLPs (*Figure 3—figure supplement 1*).

## Spatial relationships between EBOV VP40 and other viral proteins

We next analyzed the spatial relationship between eVP40 and the other viral components NC or GP. To do this, we first required the positions and orientations of the other viral components. For EBOV NP-VP24-VP35-VP40, NC positions had been calculated previously while determining the structure of the NC (*Wan et al., 2017*). For EBOV VP40-GP VLPs, we determined a low-resolution structure of Ebola GP, thereby determining its position (*Figure 5—figure supplement 1*). We then generated neighbor density maps: these show the relative distribution of all subtomograms of interest (those containing GP or NC) with respect to all reference subtomograms (those containing VP40) (*Figure 5*).

We found that the NC layer is positioned at a consistent radial distance from VP40, but otherwise shows no defined spatial relationships with VP40 particles (*Figure 5*). This arrangement is consistent with the presence of a tether (likely contributed by NP) that radially links VP40 and the NC layer. We estimated the stoichiometric ratio between VP40 and NC as ~ 4.4, suggesting that only a minority of VP40 molecules can be simultaneously bound by such a tether, and explaining the absence of any density corresponding to a bound tether in our VP40 structure.

We found that GP does not form an ordered lattice on the membrane. There is, however, a consistent radial distance between VP40 and nearby GP (*Figure 5*); this distance is related to the thickness of the membrane envelope as is expected for proteins bound on opposite sides of the envelope. A tangential view of the neighbor plot shows weak striations in the GP layer, suggesting a tendency for GP to sit in preferred positions relative to the underlying VP40. We radially projected the GP neighbor positions onto the underlying VP40 lattice, revealing that GP is preferentially located at positions near the CTD-CTD interfaces of VP40.

## Discussion

VP40 has been observed to adopt a number of different oligomeric states, including dimers, hexamers, octameric rings, and higher-order oligomers, which involve distinct conformations and assembly surfaces. Dimers are formed via interactions between VP40 NTDs, and represent the basic solubilized conformation of VP40 (*Bornholdt et al., 2013*). Octameric rings are formed using a different set of NTD-NTD interactions and appear to serve an essential RNA-binding function in a different stage of the viral life cycle distinct from its primary role as a matrix protein (*Gomis-Rüth et al., 2003*; *Hoenen et al., 2005*; *Bornholdt et al., 2013*), and a range of oligomers has been observed at the plasma membrane of infected cells (*Adu-Gyamfi et al., 2012*).

The previous model for the arrangement of VP40 within filoviruses was based upon a crystal structure obtained in the presence of dextran sulfate in which VP40 forms a hexamer. In this conformation, 6 VP40 NTDs form a linear oligomer, bracketed by a CTD on each end, with central CTDs 'sprung' and therefore disordered on each side of the linear core (*Figure 1—figure supplement 1*). In this model, the sprung CTDs protruding from one side of the NTD hexamer bind NC while those on the other side bind the plasma membrane. Higher order oligomerization of hexamers via CTD interactions were then proposed to form a matrix lattice with dimensions similar to repeating features observed in cryo-electron tomograms (*Bornholdt et al., 2013*).

The VP40 matrix structures observed here in VLPs and virions reveal a linear arrangement of VP40 dimers without sprung CTDs. We suggest that the assembly of the half-sprung hexamer could have been the result of crystal packing and/or the presence of dextran sulfate. The interactions of the central VP40s in the hexamer are similar to those seen in nucleic-acid-binding octameric VP40 rings. It is possible that dextran sulfate is not acting as a membrane mimic, but instead as a nucleic acid mimic and inducing a conformation related to nucleic-acid binding octameric VP40 rings. We therefore conclude that dimers and higher order oligomers of dimers, are the oligomeric states that play a role during virus assembly.

In all VP40 containing VLPs we studied, as well as in authentic MARV virions, the matrix layer is composed of linear chains of VP40 dimers, in which the dimeric interface is provided by the NTD, and the inter-dimer interface by the CTD. This linear arrangement of dimers is more similar to the packing of VP40 within C2 crystals and the $P6_2$ and $P6_422$ crystals presented here. In this

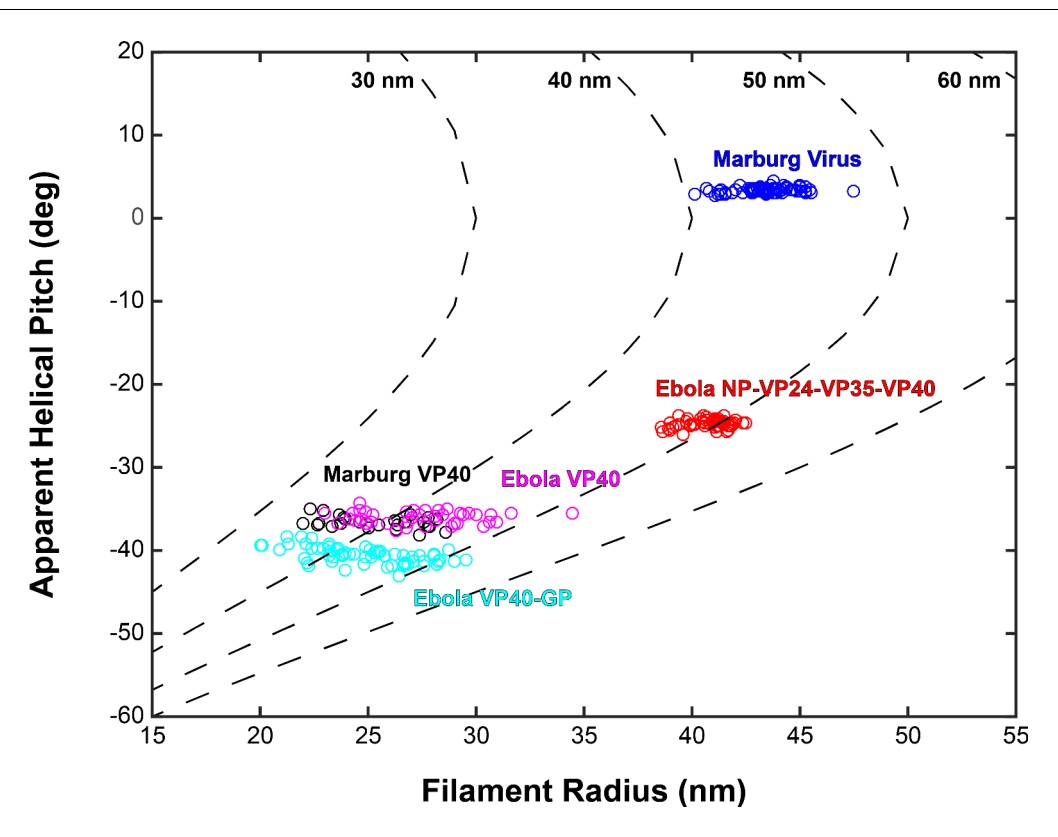

**Figure 4.** Plots of filament radii with respect to apparent helical pitch of linear VP40 chains. Scatter points represent measurements for individual filaments. Dotted lines represent expected helical pitch for given radius, assuming a constant radius of curvature. The radii of curvature plotted from left to right are 30, 40, 50, and 60 nm.

arrangement, VP40 interacts with the membrane via basic patches in the CTD. VP40 chains are stacked to form 2D lattices on the underside of the viral membrane (*Figure 6*).

The arrangement of VP40 which we observe provides a structural explanation for the phenotypes of a number of previously characterized EBOV and MARV VP40 mutants. Mutations in the basic patch which mediates the interaction between VP40 and the membrane, inhibit membrane binding, matrix assembly and budding for both eVP40 (*Bornholdt et al., 2013*) and mVP40 (*Koehler et al., 2018*). Mutations that disrupt the NTD-NTD dimeric interface prevent membrane binding, assembly, and budding for both eVP40 (*Bornholdt et al., 2013*; *Oda et al., 2016*) and mVP40 (*Koehler et al., 2018*), consistent with the key role of this interface in higher-order oligomerization of VP40. Mutations such as eVP40-M241R, which disrupts the hydrophobic patch of the CTD-CTD interface, lead to crystal forms which poorly recapitulate the CTD-CTD interface, while expression of eVP40 M241R or I307R block matrix assembly and budding (*Bornholdt et al., 2013*). Introduction of M305F/I307F instead of I307R, to enhance the hydrophobic interface, also enhances proportion of VLP release relative to expression level. Both mutants are consistent with a role for linear oligomerization of VP40 dimers via the hydrophobic CTD interface in promoting membrane curvature and filament growth (*Bornholdt et al., 2013*). Complete disruption of the CTD-CTD interface (eVP40-R134A/I307R) allows for membrane binding but not oligomerization, and neither budding nor ruffling is observed (*Bornholdt et al., 2013*). Other mutations to the CTD further disrupt membrane interactions and assembly (*Adu-Gyamfi et al., 2013*).

The matrix layer we observe in both VLPs and in MARV virions has only local order. Patches of ordered VP40 are separated by various defects in the 2D crystallographic packing. It has been suggested that VP40 VLPs elongate perpendicularly from the plasma membrane (*Kolesnikova et al., 2007b*; *Kolesnikova et al., 2007a*). While it is possible to envisage filament protrusion as mediated by highly processive extension of VP40 chains at the base of an extending filament, our data are

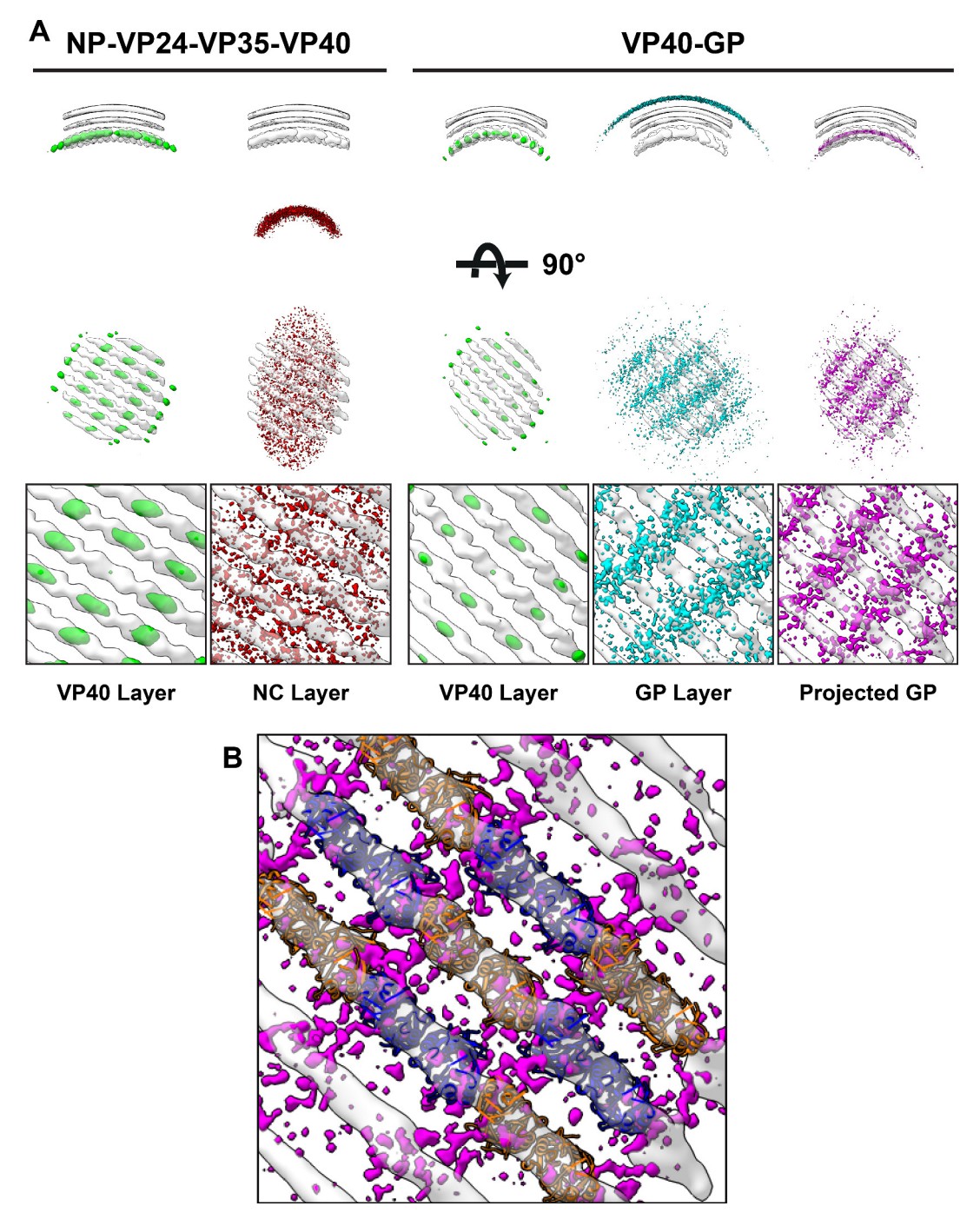

**Figure 5.** Neighbor density maps of Ebola VLPs. (**A**) First two columns are from NP-VP24-VP35-VP40 VLPs while last three columns are from VP40-GP VLPs, respective subtomogram averaging structures are shown as transparent densities. Each neighbor density map is shown as a colored isosurface indicating the preferred location of the named protein relative to the VP40 positions. Top row shows cross-sectional views through the filaments, center row shows view from outside the filaments, and bottom row shows detailed views from center row. In center and bottom rows, membrane is removed from subtomogram averages for easier viewing. The projected GP layer contains the same data as the GP layer, but projected on to the VP40 radius along the direction of the GP stalks. (**B**) A zoomed-in view of the low projected GP panel, showing the preferred positions of GP relative to a model of the VP40 layer.

The online version of this article includes the following figure supplement(s) for figure 5:

**Figure supplement 1.** Structure of Ebola GP from eVP40-GP VLPs.

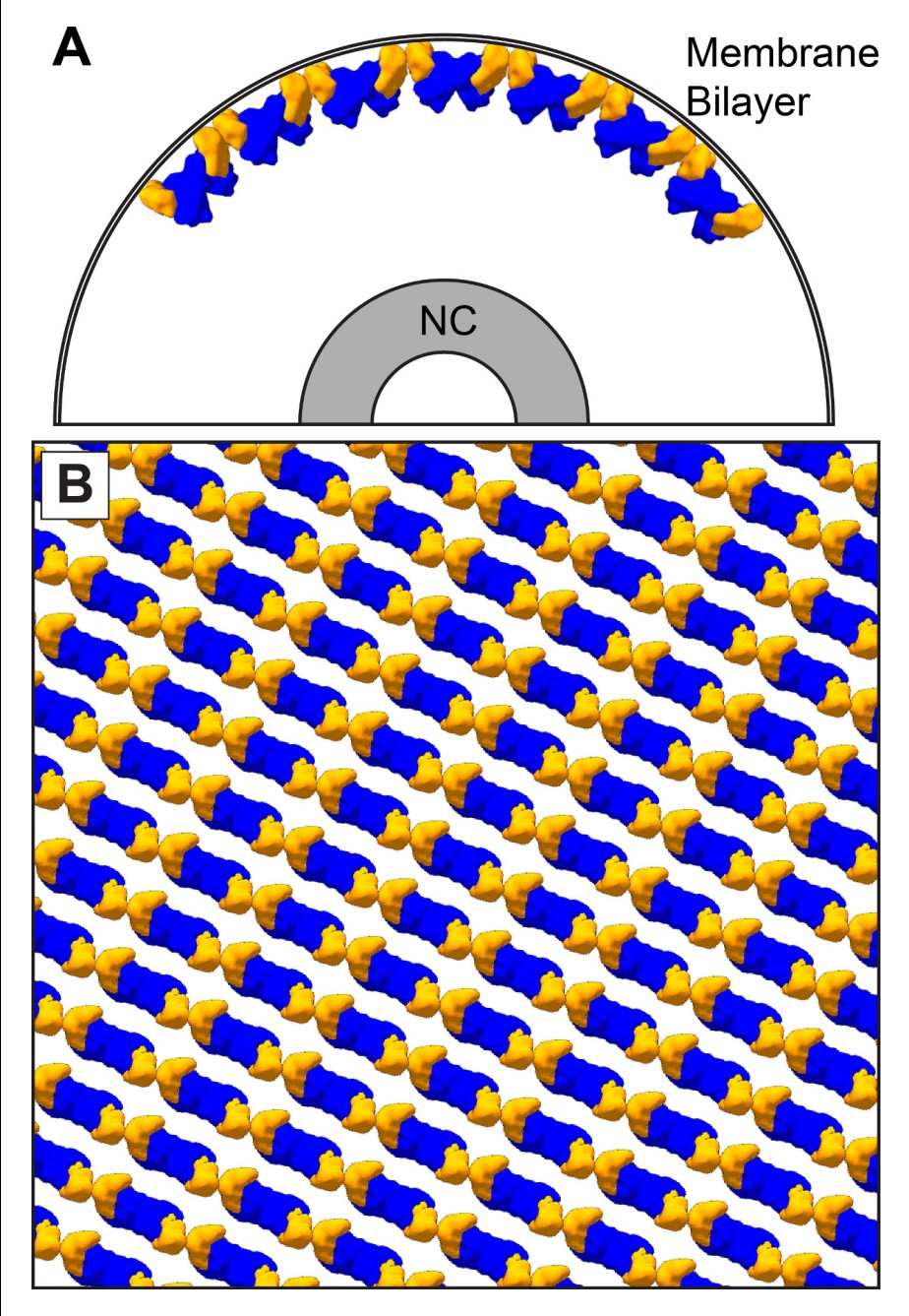

**Figure 6.** The dimer-chain matrix model. VP40 NTDs are shown in blue and CTDs are shown in orange. Our dimer-chain model, where VP40 dimers form linear chains directly below the membrane surface and an unknown non-VP40 component tethers the NC to the matrix layer. (A) Shows a cross-section view, while (B) Shows a surface view directly below the membrane surface.

more consistent with assembly of the VP40 lattice from multiple starting points to generate a patchwork of locally ordered lattices at the assembly site. This mode of assembly is also easier to reconcile with our previous data suggesting that the budding of virions containing an NC takes place like a surfacing submarine, where the NC initially protrudes parallel to the membrane surface before being wrapped from one end (*Welsch et al., 2010*). Because VP40 chains adopt preferred curvatures and therefore are oriented in a defined orientation relative to the membrane, neighboring patches would tend to agglomerate with their VP40 chains in approximately the same orientation.

We were unable to determine the structure of VP40 in Ebola virions, as the majority of viruses showed unstructured matrix layers. We think this is likely a fixation artefact, as Ebola virus preparations were generally more delicate than Marburg virus preparations. However, given our results with VLPs and the observations of ordered patches in some Ebola virions, we suggest that VP40 forms patchwork lattices of the same structure in Ebola virus.

We do not observe any substantial contribution to matrix density from another protein. However, given the $\sim$ 4.4 VP40:NP ratio which we observed, we cannot rule out that a small part of every NP binds to one VP40 molecule, since the sub-stoichiometric levels of binding of a small additional density might not be detected.

GP has been previously shown to migrate toward VP40-rich membrane areas and colocalize with VP40 in VLPs (*Licata et al., 2004*; *Noda et al., 2002*). We observed that in EBOV VP40-GP VLPs, GP has a tendency to locate to striations that run perpendicular to the VP40 chains. These data suggest that GP preferentially sits at positions close to the inter-dimer CTD-CTD interfaces. Such a preferential localization could be derived through a direct interaction between VP40 CTD and the short, five-residue cytoplasmic tail of GP. Alternatively, VP40 CTD may modify the local lipid composition to generate a local environment favorable to the GP trans-membrane domain.

Our data reveal the arrangement of VP40 in assembled filovirus particles. They are consistent with a model for filovirus assembly in which VP40 dimers in solution migrate toward the plasma membrane, where they oligomerize into curved chains via CTD-CTD interactions, which induces local membrane curvature. Stacking of VP40 chains results in the formation of 2D lattices which are curved in one direction. Membrane curvature can be propagated over larger areas of the membrane by growth of patches of 2D lattice or by contact and 'fusion' between neighboring patches.

## Data deposition

EM maps of VP40 from Ebola NP-VP24-VP35-VP40, VP40, VP40-GP VLPs and Marburg virions and VP40 VLPs were deposited in the EMDB with accession numbers EMD-11660, EMD-11661, EMD-11662, EMD-11663, EMD-11664, respectively. EM map of Ebola GP was deposited as EMD-11665. Crystal structures of eVP40 $P6_2$ and eVP40 $P6_422$ were deposited to the PDB with accession numbers 7JZJ and 7JZT, respectively.

# Materials and methods

**Key resources table**

| Reagent type (species) or resource | Designation | Source or reference | Identifiers | Additional information |
|---|---|---|---|---|
| Strain, strain background (*Escherichia coli*) | BL21(DE3) | Novagen Rosetta 2 (DE3) Merck (Darmstadt, Germany) | | Competent cells |
| Cell line (*Homo-sapiens*) | HEK-293T | American Type Culture Collection | | VLP production |
| Cell line (*Homo-sapiens*) | Huh7 | Japanese Collection of Research Bioresources | | Marburg virus production |
| Recombinant DNA reagent | pCAGGS-ZEBOV-NP (plasmid) | *Hoenen et al., 2006* | | |
| Recombinant DNA reagent | pCAGGS-ZEBOV-VP24 (plasmid) | *Hoenen et al., 2006* | | |
| Recombinant DNA reagent | pCAGGS-ZEBOV-VP35 (plasmid) | *Hoenen et al., 2006* | | |
| Recombinant DNA reagent | pCAGGS-ZEBOV-VP40 (plasmid) | *Hoenen et al., 2006* | | |
| Recombinant DNA reagent | pCAGGS-ZEBOV-GP (plasmid) | *Hoenen et al., 2006* | | |

*Continued on next page*

*Continued*

| Reagent type (species) or resource | Designation | Source or reference | Identifiers | Additional information |
|---|---|---|---|---|
| Recombinant DNA reagent | pCAGGS-MARV-VP40 (plasmid) | *Wenigenrath et al., 2010* | | |
| Recombinant DNA reagent | pTriex5-Strep-zVP40-WT | *Bornholdt et al., 2013* | | |
| Recombinant DNA reagent | pTriex5-Strep-zVP40-R134A/I307R | *Bornholdt et al., 2013* | | |
| Recombinant DNA reagent | pTriex5-Strep-zVP40-I307F | this manuscript | | Plasmid construction is described in materials and methods |
| Recombinant DNA reagent | pTriex5-Strep-zVP40-M305F/I307F | this manuscript | | Plasmid construction is described in materials and methods |
| Recombinant DNA reagent | pET46+ zVP40-d43 | *Bornholdt et al., 2013* | | |
| Virus | Marburg Virus | | | |
| Software, algorithm | autoPROC | *Vonrhein et al., 2018* | | |
| Software, algorithm | PHENIX | *Adams et al., 2010* | | |
| Software, algorithm | COOT | *Emsley et al., 2010* | | |
| Software, algorithm | SerialEM | *Mastronarde, 2005* | | |
| Software, algorithm | MotionCorr | *Li et al., 2013* | | |
| Software, algorithm | CTFFIND4 | *Rohou and Grigorieff, 2015* | | |
| Software, algorithm | ctfphaseflip | *Xiong et al., 2009* | | |
| Software, algorithm | IMOD | *Kremer et al., 1996* | | |
| Software, algorithm | Amira | Thermo Fisher Scientific | | |
| Software, algorithm | EM-toolbox | *Pruggnaller et al., 2008* | | |
| Software, algorithm | TOM-toolbox | *Nickell et al., 2005* | | |
| Software, algorithm | AV3 | *Förster et al., 2005* | | |
| Software, algorithm | dynamo | *Castaño-Díez et al., 2012* | | |
| Software, algorithm | UCSF Chimera | *Pettersen et al., 2004* | | |

## Expression, crystallization, and crystal structure determination of Ebola VP40

eVP40 was expressed in *E. coli* BL21 cells as previously described (*Bornholdt et al., 2013*) and crystallized in 100 mM HEPES, 50 mM MgCl2, 38% PEG400, pH 7.2. Crystals belonging to the $P6_2$ space group diffracted to 2.4 Å, at Beamline 12–2 of the Stanford Synchrotron Radiation Lightsource

(SSRL). Crystals belonging to the space group P6$_4$22 space group diffracted anisotropically to 3.7 Å, at the Argonne National Laboratory, Beamline SBC- 19-ID. Data integration and scaling were performed using the autoPROC implementation of XDS and AIMLESS (*Vonrhein et al., 2011*). Anisotropy correction of the eVP40 P6$_4$22 data set was performed using STARANISO with a surface threshold of 1.2/σ(I), implemented through the autoPROC pipeline (*Vonrhein et al., 2018*). Isotropic data were used for model building and refinement of the eVP40 P6$_2$ crystal form and anisotropic corrected data were used for model building and refinement of the eVP40 P6$_4$22 crystal form. Both structures were determined using molecular replacement using PHENIX (*Adams et al., 2010*) with dimeric eVP40 (PDB: 4LDB) as the search model. Refinement of each crystal structure was done through iterative rounds of manual model building using COOT (*Emsley et al., 2010*), followed by refinement of the models in PHENIX.

## Cell lines

HuH7 cells were obtained from Japanese Collection of Research Bioresources and tested for mycoplasma contamination. HEK293T cells were obtained from American Type Culture Collection. They are listed in the database of commonly misidentified cell lines maintained by the International Cell Line Authentication Committee, but were used as they are well-established tools for the expression of VLPs: the cells themselves were not studied.

## Expression and purification of VLPs

HEK293T cells were transfected with the appropriate combination of full length plasmids in pCAGGS backbones: full-length Marburg virus VP40; Zaire Ebola virus VP40; Zaire Ebola virus VP40 and GP; or Zaire Ebola virus NP, VP24, VP35, and VP40 (*Hoenen et al., 2006*; *Wenigenrath et al., 2010*). Supernatant was collected 3 days after transfection and clarified by centrifugation at 800 g for 10 min at 4°C. The remaining steps were performed at 4°C. VLPs were pelleted through a 20% (w/v) sucrose cushion in TNE buffer (50 mm Tris–HCl pH 7.4, 100 mm NaCl, 0.1 mm EDTA) at 160,000 g for 3 hr, resuspended in TNE buffer, and separated on a Nycodenz step gradient (2.5%, 5%, 7.5%, 10%, 15%, 20%, 30% (v/v)) at 34,400 g for 15 min. Fractions 4–6 were collected and checked by negative-stain EM; fractions confirmed to contain VLPs were pooled and pelleted at 92,000 g for 2 hr. Final pellets were resuspended in TNE.

## VLP budding assay

Budding of virus-like particles (VLPs) into cell supernatants was detected by Western blot analyses. Wild-type and mutant VP40 bearing a Strep-Tag were cloned into pTriEx-5 (Novagen) and transfected into cells using TrasnIT-LT1 transfection reagent (Mirus). VLPs were harvested 24 hr post-transfection. Cell culture medium was spun down at 3500 rpm for 20 min to pellet any cells out of the media. The cleared supernatants were then ultracentrifuged at 30,000 rpm with an SW-60 rotor (Beckman) for 2 hr through a 20% (w/v) sucrose cushion-50 mM Tris pH 7.4, 100 mM NaCl. Pelleted VLPs were resuspended in 1X NuPAGE LDS sample buffer (ThermoFisher). Cell lysates were collected by washing cells twice with PBS followed by lysis in CytoBuster. VLPs and cell lysates were then run on SDS denaturing gels, transferred onto polyvinylidene difluoride (PVDF) Immobilon transfer membranes (Millipore), and probed with an anti-Strep-Tag antibody (GeneTex). The relative intensities of the bands were quantified by densitometry with a ChemiDoc MP imaging system (Bio-Rad) and ImageJ. The budding index was defined as the amount of Strep-VP40 in the VLPs divided by the amount in the cell lysate and presented as % of wild-type Strep-VP40.

## Preparation of inactivated Marburg virus

Virus specimens were grown, purified, and fixed under BSL-4 conditions as previously described (*Bharat et al., 2011*). Briefly, Huh7 cells were infected with Marburg virus. Supernatant was collected 1 day post infection, and centrifuged at 4°C for 2 hr at approximately 77,000 g through a 20% (w/v) sucrose cushion to isolate the virus particles. The resultant virus pellet was resuspended in calcium and magnesium deficient phosphate-buffered saline (PBS), re-pelleted, and inactivated with paraformaldehyde in DMEM (final concentration 4%) for 24 hr by filling the tube completely. The viruses were pelleted and the 4% paraformaldehyde solution in DMEM (w/v) was replaced with a fresh

solution of 4% paraformaldehyde. The sample was released from the BSL-4 facility after an additional 24 hr.

## Cryo-electron tomography

C-Flat 2/2–3C grids stored under vacuum were glow discharged for 30 s at 20 mA. Virus or VLP solution was diluted with 10 nm colloidal gold; 2.5 µl of this mixture was applied to each grid and plunge frozen into liquid ethane using a FEI Vitrobot Mark 2. Grids were stored in liquid nitrogen until imaging.

Tomographic imaging was performed as described previously (*Schur et al., 2016*; *Wan et al., 2017*). Briefly, imaging was performed on a FEI Titan Krios at 300 keV using a Gatan Quantum 967 LS energy filter with a slit width of 20 eV and a Gatan K2xp detector in super-resolution mode. Tomograms were acquired from −60° to 60° with 3° steps using SerialEM (*Mastronarde, 2005*) and a scripted dose-symmetric tilt-scheme (*Hagen et al., 2017*). Data collection parameters are provided in *Table 3*.

Frames were aligned with either K2Align software, which uses the MotionCorr algorithm (*Li et al., 2013*), or with the frame alignment algorithm built into serialEM; aligned frames were Fourier cropped to 4k × 4 k, giving a final pixel size of 1.78 Å per pixel. Defocus for each tilt was determined by CTFFIND4 (*Rohou and Grigorieff, 2015*). Tilt images were filtered by cumulative electron dose using the exposure-dependent attenuation function and critical exposure constants as described elsewhere (*Schur et al., 2016*).

Contrast transfer functions (CTFs) of individual images were corrected using ctfphaseflip (*Xiong et al., 2009*) and tomograms were reconstructed using weighted back projection in IMOD (*Kremer et al., 1996*). Tomograms with poor fiducial alignment were discarded; poor fiducial alignment was defined as alignment residual above one pixel in 2 × binned data or retaining fewer than eight fiducial markers. CTF-corrected unbinned tomograms were binned by 2× (3.56 Å per pixel) and 4× (7.12 Å per pixel) by Fourier cropping.

## Subtomogram averaging

Filaments of interest were first identified in 4×-binned tomograms using Amira visualization software (FEI Visualization Sciences Group). Using Amira and the electron microscopy toolbox (*Pruggnaller et al., 2008*), points were selected along the central filament axes and radii were measured along the matrix layers. These were then used to define the filament axes and generate an oversampled cylindrical grid for each filament along the matrix layer. These gridpoints served as

**Table 3.** Data collection and image processing table.

| | EBOV NP-VP24-VP35-VP40 (EMD-11660) | EBOV VP40 (EMD −11661) | EBOV VP40-GP (EMD −11662) | MARV (EMD −11663) | MARV VP40 (EMD −11664) |
|---|---|---|---|---|---|
| Magnification | 81,000x | 81,000x | 81,000x | 81,000x | 81,000x |
| Voltage (kV) | 300 | 300 | 300 | 300 | 300 |
| Electron exposure (e-/ Å$^2$) | ~100 | ~100 | ~100 | ~80 | ~100 |
| Defocus range (µm) | −2.0 to −4.5 | −2.0 to −4.5 | −2.0 to −4.5 | −2.0 to −4.5 | −2.0 to −4.5 |
| Detector | Gatan Quantum K2 | Gatan Quantum K2 | Gatan Quantum K2 | Gatan Quantum K2 | Gatan Quantum K2 |
| Energy filter | Yes | Yes | Yes | Yes | Yes |
| Slit width (eV) | 20 | 20 | 20 | 20 | 20 |
| Tilt Range (min/max, step) | −60°/60°, 3° | −60°/60°, 3° | −60°/60°, 3° | −60°/60°, 3° | −60°/60°, 3° |
| Pixel Size (Å) | 1.78 | 1.78 | 1.78 | 1.78 | 1.78 |
| Tomograms (used/acquired) | 52/64 | 39/42 | 55/73 | 76/82 | 34/35 |
| Filaments | 54 | 43 | 65 | 93 | 34 |
| Symmetry | C2 | C2 | C2 | C2 | C2 |
| Final subtomograms (no.) | 59580 | 20352 | 106793 | 75212 | 42938 |
| Map resolutions (FSC = 0.143) | 10.2 Å | 9.8 Å | 9.9 Å | 9.6 Å | 10.8 Å |

initial extraction points for subtomograms. Initial Euler angles for each gridpoint were derived from the cylindrical grid. These initial positions and orientations were used to generate the initial motive-list, the metadata file for subtomogram averaging.

Initial references were generated by subtomogram averaging of single filaments using 4 × binned data. Subtomogram averaging was performed using TOM (*Nickell et al., 2005*), AV3 (*Förster et al., 2005*) and dynamo (*Castaño-Díez et al., 2012*), and scripts derived from their functions. Using the initial motivelist, the initial average that was roughly a cylindrically averaged section of a filament. From there a six-dimensional search was performed to refine Euler angles and Cartesian shifts, resulting in a low-resolution structure.

At this point, it became clear that the matrix layer was not helical in structure and had C2 symmetry, indicating the structures were apolar with respect to the filament axis. As such, initial references for each speciment were used to align the full datasets using C2 symmetry. Initial alignments were performed using 4 × binned data and a low pass filter limiting resolutions to 35 Å. After convergence of subunit positions, oversampled particles were removed by distance thresholding. Each tomogram was also thresholded by cross-correlation to exclude subtomograms that had misaligned to positions away from the matrix layer. The unique particle parameters were then split into 'odd and even' sets, and aligned independently from this point on. Subtomograms were re-extracted with 2 × binning and halfsets were aligned independently until the six-dimensional search converged. This was then repeated with 1 × binned data.

Final resolutions were measured using a mask-corrected FSC (*Chen et al., 2013*), and final averages were low-pass filtered, sharpened, CTF-reweighted, and figure-of-merit weighted to their determined resolutions as previously described (*Schur et al., 2016*). Data processing parameters are provided in *Table 3*.

## Visualization and rigid body fitting

Visualization of tomograms and electron density maps were done with University of California, San Francisco (UCSF) Chimera (*Pettersen et al., 2004*). Rigid body fitting of atomic models into density maps was performed using the fit-in-map function in UCSF Chimera.

## Measuring 2D crystal lattices

Approximate 2D crystal lattices were measured from the subtomogram averages. Prior to measurement, the structures were 'unwrapped' from Cartesian space to cylindrical polar space, allowing for direct measurement along the cylindrical surface. Measurements were performed near the middle of the VP40 dimeric interface.

## Lattice maps and neighbor density plots

The data for lattice maps are the positions and the orientations of the subtomograms determined during subtomogram averaging. Lattice maps were visualized in UCSF Chimera using the Place Objects plugin (*Qu et al., 2018*).

Neighbor density plots are calculated by first picking a reference subtomogram, then finding all neighbors within a given distance threshold. The reference subtomogram, along with its neighbors, is then shifted and rotated into the center of the density plot, and all neighbors are added to the plot. When performed across all subtomograms, the result is a set of point clouds that represent the probability of finding a neighboring subtomogram in those positions. The probability distributions of the point cloud should reflect the positions of subunits in the subtomogram averages, with neighbor density clouds becoming more dispersed away from the center of the plot, reflecting the loss of resolution away from the center of the average.

Cross-neighbor density maps are calculated using two motivelists, with one containing the reference subtomograms, and the other containing the orientations of the second proteins of interest.

## Acknowledgements

The Briggs laboratory acknowledges financial support from the European Molecular Biology Laboratory, the Medical Research Council (MC_UP_1201/16) and the European Research Council (ERC) under the European Union's Horizon 2020 research and innovation programme (ERC-CoG-648432 MEMBRANEFUSION). The Becker group was supported by the Deutsche Forschungsgemeinschaft

(Sonderforschungsbereich 1021) and by the German Center for Infection Research (DZIF). The Saphire group was supported by institutional funds of the La Jolla Institute for Immunology. This work was supported by an EMBO long-term fellowship, ALTF 748–2014, awarded to WW. We thank A Tan for assistance with preliminary data processing, DM Abelson for assistance with mutagenesis and W J H Hagen (EMBL Heidelberg) for assistance during tomographic data collection. The SSRL Structural Molecular Biology Program is supported by the DOE Office of Biological and Environmental Research, and by the National Institutes of Health, National Institute of General Medical Sciences (including P41GM103393). The Advanced Photon Source is a U.S. Department of Energy (DOE) Office of Science User Facility operated for the DOE Office of Science by Argonne National Laboratory under Contract No. DE-AC02-06CH11357.

## Additional information

### Funding

| Funder | Grant reference number | Author |
| --- | --- | --- |
| Medical Research Council | MC_UP_1201/16 | John AG Briggs |
| H2020 European Research Council | ERC-CoG-648432 | John AG Briggs |
| Deutsche Forschungsgemeinschaft | Sonderforschungsbereich 1021 | Stephan Becker |

The funders had no role in study design, data collection and interpretation, or the decision to submit the work for publication.

### Author contributions

William Wan, Conceptualization, Formal analysis, Investigation, Visualization, Methodology, Writing - original draft, Writing - review and editing; Mairi Clarke, Zachary A Bornholdt, Investigation; Michael J Norris, Formal analysis, Investigation, Visualization, Writing - original draft, Writing - review and editing; Larissa Kolesnikova, Resources, Writing - review and editing; Alexander Koehler, Resources; Stephan Becker, Conceptualization, Supervision, Funding acquisition, Project administration, Writing - review and editing; Erica Ollmann Saphire, Conceptualization, Supervision, Funding acquisition, Writing - original draft, Project administration; John AG Briggs, Conceptualization, Supervision, Funding acquisition, Methodology, Writing - original draft, Project administration, Writing - review and editing

### Author ORCIDs

William Wan http://orcid.org/0000-0003-2497-3010
Mairi Clarke http://orcid.org/0000-0002-9658-4308
Michael J Norris https://orcid.org/0000-0002-8325-5257
Zachary A Bornholdt http://orcid.org/0000-0002-7557-9219
Erica Ollmann Saphire https://orcid.org/0000-0002-1206-7451
John AG Briggs https://orcid.org/0000-0003-3990-6910

### Decision letter and Author response

Decision letter https://doi.org/10.7554/eLife.59225.sa1
Author response https://doi.org/10.7554/eLife.59225.sa2

## Additional files

### Supplementary files
- Transparent reporting form

## Data availability

EM maps of VP40 from Ebola NP-VP24-VP35-VP40, VP40, VP40-GP VLPs and Marburg virions and VP40 VLPs were deposited in the EMDB with accession numbers EMD-11660, EMD-11661, EMD-11662, EMD-11663, EMD-11664, respectively. EM map of Ebola GP was deposited as EMD-11665. Crystal structures of eVP40 P62 and eVP40 P6422 were deposited to the PDB with accession numbers 7JZJ and 7JZT, respectively.

The following datasets were generated:

| Author(s) | Year | Dataset title | Dataset URL | Database and Identifier |
|---|---|---|---|---|
| Wan W, Clarke M, Norris M, Kolesnikova L, Koehler A, Bornholdt ZA, Becker S, Saphire EO, Briggs JAG | 2020 | Crystal structures of eVP40 P62 | https://www.rcsb.org/structure/7JZJ | RCSB Protein Data Bank, 7JZJ |
| Wan W, Clarke M, Norris M, Kolesnikova L, Koehler A, Bornholdt ZA, Becker S, Saphire EO, Briggs JAG | 2020 | Crystal structures of eVP40 P6422 | https://www.rcsb.org/structure/7JZT | RCSB Protein Data Bank, 7JZT |
| Wan W, Clarke M, Norris M, Kolesnikova L, Koehler A, Bornholdt ZA, Becker S, Saphire EO, Briggs JAG | 2020 | EM map of VP40 from Ebola NP-VP24-VP35-VP40 | http://www.ebi.ac.uk/pdbe/entry/emdb/EMD-11660 | Electron Microscopy Data Bank, EMD-11660 |
| Wan W, Clarke M, Norris M, Kolesnikova L, Koehler A, Bornholdt ZA, Becker S, Saphire EO, Briggs JAG | 2020 | EM map from VP40 | http://www.ebi.ac.uk/pdbe/entry/emdb/EMD-11661 | Electron Microscopy Data Bank, EMD-11661 |
| Wan W, Clarke M, Norris M, Kolesnikova L, Koehler A, Bornholdt ZA, Becker S, Saphire EO, Briggs JAG | 2020 | EM map from VP40-GP VLP | http://www.ebi.ac.uk/pdbe/entry/emdb/EMD-11662 | Electron Microscopy Data Bank, EMD-11662 |
| Wan W, Clarke M, Norris M, Kolesnikova L, Koehler A, Bornholdt ZA, Becker S, Saphire EO, Briggs JAG | 2020 | EM map from Marburg virions | http://www.ebi.ac.uk/pdbe/entry/emdb/EMD-11663 | Electron Microscopy Data Bank, EMD-11663 |
| Wan W, Clarke M, Norris M, Kolesnikova L, Koehler A, Bornholdt ZA, Becker S, Saphire EO, Briggs JAG | 2020 | EM map from VP40 VLPs | http://www.ebi.ac.uk/pdbe/entry/emdb/EMD-11664 | Electron Microscopy Data Bank, EMD-11664 |
| Wan W, Clarke M, Norris M, Kolesnikova L, Koehler A, Bornholdt ZA, Becker S, Saphire EO, Briggs JAG | 2020 | EM map of Ebola GP | http://www.ebi.ac.uk/pdbe/entry/emdb/EMD-11665 | Electron Microscopy Data Bank, EMD-11665 |

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
