## [Decision Letter]

**Acceptance summary:**

Filoviruses like Ebola and Marburg are important human pathogens, and they have fascinating filamentous virion structures that are organized by the VP40 protein. The manuscript by Briggs and colleagues now describes the molecular architecture of VP40 in Ebola and Marburg filovirus-like particles and in fixed Marburg virus particles. The work nicely resolves the structural organization of VP40 within virus particles, which has up to now been unclear because multiple different VP40 interactions and conformations have been postulated or observed in crystal lattices. The current analysis shows that VP40 forms dimers that interact with one other via their C-terminal domains, to form extended filaments. This arrangement exposes two basic patches for membrane binding, the filaments are slightly bent (and somewhat flexible), which explains how VP40 bend membranes (and at different radii) during budding, and the GP protein anchors preferentially at the CTD-CTD interaction sites.

**Decision letter after peer review:**

Thank you for submitting your article "Ebola and Marburg virus matrix layers are locally ordered assemblies of VP40 dimers" for consideration by *eLife*. Your article has been reviewed by two peer reviewers, and the evaluation has been overseen by a Reviewing Editor and Cynthia Wolberger as the Senior Editor. The following individual involved in review of your submission has agreed to reveal their identity: Eileen Jaffe (Reviewer #3).

The reviewers have discussed the reviews with one another and the Reviewing Editor has drafted this decision to help you prepare a revised submission.

Summary:

The manuscript by Briggs and colleagues describes the molecular architecture of VP40 in Ebola and Marburg filovirus-like particles and in fixed Marburg virus particles. The work nicely reveals the structure and organization of VP40 within virus particles. VP40 forms dimers that interact with one other via their C-terminal domains, to form extended filaments. This arrangement exposes two basic patches for membrane binding, the filaments are slightly bent (and somewhat flexible), which explains how VP40 bend membranes (and at different radii) during budding, and the GP protein anchors preferentially at the CTD-CTD interaction sites.

This is excellent work and it provides important new insights into the role of VP40 in filovirus assembly and budding.

Suggestions for the authors’ consideration:

1) It is not clear that anything is gained by showing the incorrect model for VP40 assembly and membrane interaction in Figure 6A (and it might even be confusing to some readers). We recommend that the authors consider removing that model.

2) It would be helpful if the authors could describe explicitly in the Discussion how they view the relationship between the VP40 filament structure described in this manuscript and the other reported conformers of VP40 (e.g., the dimer and octamer, and perhaps also the different kinds of hexamers that have been reported previously). One of the interesting aspects of VP40 is its ability to adopt different oligomeric states that (apparently) perform different functions, and we recommend making sure that this comes through clearly in the Discussion (i.e., which structures have clear biological roles?, which structures lack clear roles?, where are the uncertainties?)

---

## [Author Response]

Suggestions for the authors’ consideration:1) It is not clear that anything is gained by showing the incorrect model for VP40 assembly and membrane interaction in Figure 6A (and it might even be confusing to some readers). We recommend that the authors consider removing that model.

We have followed the recommendation and removed the model.

2) It would be helpful if the authors could describe explicitly in the Discussion how they view the relationship between the VP40 filament structure described in this manuscript and the other reported conformers of VP40 (e.g., the dimer and octamer, and perhaps also the different kinds of hexamers that have been reported previously). One of the interesting aspects of VP40 is its ability to adopt different oligomeric states that (apparently) perform different functions, and we recommend making sure that this comes through clearly in the Discussion (i.e., which structures have clear biological roles?, which structures lack clear roles?, where are the uncertainties?).

As suggested, we have added further discussion of the different oligomeric states of VP40.